# Pruning vs Quantization: Which is Better?

**Andrey Kuzmin, Markus Nagel, Mart van Baalen, Arash Behboodi, Tijmen Blankevoort**
Qualcomm AI Research*
Amsterdam, The Netherlands
{akuzmin, markusn, mart, behboodi, tijmen}@qti.qualcomm.com

## Abstract

Neural network pruning and quantization techniques are almost as old as neural networks themselves. However, to date only ad-hoc comparisons between the two have been published. In this paper, we set out to answer the question on which is better: neural network quantization or pruning? By answering this question, we hope to inform design decisions made on neural network hardware going forward. We provide an extensive comparison between the two techniques for compressing deep neural networks. First, we give an analytical comparison of expected quantization and pruning error for general data distributions. Then, we provide lower bounds for the per-layer pruning and quantization error in trained networks, and compare these to empirical error after optimization. Finally, we provide an extensive experimental comparison for training 9 large-scale models on 4 tasks. Our results show that in most cases quantization outperforms pruning. Only in some scenarios with very high compression ratio, pruning might be beneficial from an accuracy standpoint. [1]

## 1   Introduction

Recent advances in deep learning led to exceeding human-level performance in many tasks, including computer vision, machine translation, voice recognition, and language understanding. Real-world applications of DNNs rely heavily on their efficiency. Both mobile and cloud platforms greatly benefit from reduced latency and energy efficiency achieved by some form of model compression. In this work, we consider two mainstream techniques used in practice; pruning and quantization.

Pruning methods remove individual weights [70, 25], or sometimes groups of weights [28, 47]. This procedure can reduce the memory footprint. Furthermore, not having to perform the computations with weights that are zeroed out can make network inference more efficient. On the other hand, quantization reduces the bit-width used for both the weights and the computation used in networks, leading to both predictable memory savings and reductions in the necessary compute. In both scenarios, the hardware used for making use of these optimization schemes needs to take them into account.

Depending on the availability of training data and computing budget, most methods for pruning and quantization fall into one of two families. The first family includes fine-tuning approaches, namely quantization-aware training (QAT) and fine-tuning with pruning in the loop. The second family includes post-training approaches such as post-training quantization (PTQ). Previously, pruning techniques primarily relied on fine-tuning; however, some post-training pruning methods appeared recently as fine-tuning is not desirable for large language models [18].

Despite the importance of model efficiency and the plethora of approaches for pruning and quantization, the two fields are mostly disjoint. The literature presents little insight into which of the two

---

*   Qualcomm AI Research is an initiative of Qualcomm Technologies, Inc.

[1]Code is available at https://github.com/Qualcomm-AI-research/pruning-vs-quantization

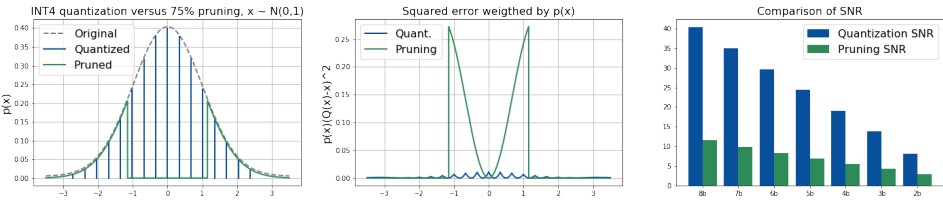

Figure 1: Comparison for a standard normal distribution. (left) Distributions after pruning and quantization for INT4 and 75% pruning. (middle) The squared error weighted by probability. (right) SNR for different compression ratios.

techniques is more accurate. In practice, there is only limited time to compress a network and limited energy to spend on making deep learning inference hardware. For this reason, we ask the question: Should one focus on quantization or pruning for compression?

We present an extensive study comparing pruning and quantization in equal settings. First, we consider different data distributions and analyze the conditions under which each method is preferable. We match our findings with real weight tensors from pre-trained models. Second, we consider a post-training scenario and evaluate single-layer output errors for both methods. Because the comparison might depend on the specific choice of optimization method, we compare the two with theoretical bounds that apply regardless of the optimization method. Finally, we provide a full-model comparison for the most common scenario of fine-tuning networks after either pruning or quantization.

In our comparison, we intentionally avoid considering the hardware aspects of pruning and quantization. Instead, we focus solely on the accuracy of both methods, given similar theoretical compression ratios. A coarse discussion on the hardware necessary for both methods can be found in section 6.

## 2   Assumptions

In our work, we assume FP16 as the basic data type and measure any gains in compression with respect to it. Using FP16 for inference generally does not lead to a loss in accuracy. Neural networks are also very commonly trained with FP16, making it a common baseline. Thus, we compare 50% pruning sparsity to INT8 quantization, 75% sparsity to INT4 quantization and so forth. We also assume no overhead on storing the sparsity mask for pruning and relegate such hardware-specific implementations to section 6.

For the pruning experiments, we consider magnitude pruning. It is common to do fine-tuning after or during pruning [70]. Several works have independently shown that despite its simplicity, it is tough to improve upon magnitude pruning and fine-tuning [19, 4]. To our knowledge, no pruning algorithm exists that consistently outperforms this method.

For the quantization experiments, we use symmetric uniform quantization, which is defined by just the quantization scale factor and the bit-width. The scale is represented as a floating-point number and is used to map floating-point values to the integer grid. Further details on symmetric uniform quantization can be found in [49]. Uniform quantization is the standard in the quantization literature, and symmetric quantization is mostly employed for the weights. In all our experiments, we use a quantization range estimator minimizing the mean-squared error on weights by grid search [49].

## 3   Comparison on statistical distributions

Before diving into comparison results, we first describe theoretically what the quantization error and pruning error are. Looking at this with a theoretical lens helps with understanding the later experimental difference between the two methods. We start off by describing and analyzing both methods on simple data distributions.

In order to compare the error of pruning and quantization, we will frequently use the signal-to-noise ratio measure defined in the log scale: $\text{SNR}_{dB} = 10 \log_{10} \left( \mathbb{E}\left[ W^2 \right] / \mathbb{E}\left[ (W - F(W))^2 \right] \right)$, where $F(W)$ is the quantization or pruning function. This measure is the same as a scaled logarithm of

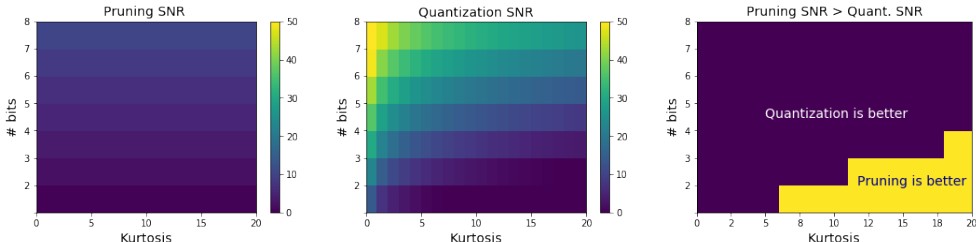

Figure 2: Comparing the error of pruning and quantization for a student-t distribution, simulating the presence of significant outliers. We plot the results for different magnitudes of the outliers, as per the kurtosis on the x-axis. (left) the pruning error, which does not change under the presence of more severe outliers. (middle) the quantization SNR, which is reduced greatly when outliers increase (right) the trade-off regions where quantization and pruning are better.

an MSE measure. Both are often employed to analyze the sensitivity of neural network layers to quantization, and they are theoretically well-founded to correlate with network performance [41, 48].

### 3.1 Quantization error

For quantization, we consider symmetric uniform quantization, which is also called integer quantization. Given a bit-width $b$ and the scale $\delta$, the grid nodes are defined as $q_i = \delta i, i \in \{-2^b, \ldots, 0, 2^b - 1\}$. The quantization operation rounding-to-nearest $Q(w)$ and the corresponding quantization error $R(w)$ are defined as:

$$Q(w) = q_i, \; i = \arg\min_i |w - q_i|, \qquad\qquad R(w) = Q(w) - w. \qquad (1)$$

Following [36] we model neural network weights as a random variable $W \sim p(w)$. The expected value of the quantization MSE can be expressed as follows:

$$\mathbb{E}\left[(Q(W) - W)^2\right] = \int\limits_{q_{min}}^{q_{max}} R^2(w)p(w)dw + \int\limits_{-\infty}^{q_{min}} (w - q_{min})^2 p(w)dw + \int\limits_{q_{max}}^{\infty} (q_{max} - w)^2 p(w)dw,$$

$$(2)$$

where $q_{min} = \min_i q_i$ and $q_{max} = \max_i q_i$ are the quantization range limits. The left term corresponds to the rounding error, and the right two terms correspond to the clipping error. We use this analytic formulation for our distribution results below, the details are given in appendix A.

### 3.2 Pruning error

We consider magnitude pruning $T(x) = x \cdot \mathbb{1}_{-t \leq x \leq t}$. This simply sets the values closest to zero to actual zero. Given this, the expected error of pruning is expressed as follows:

$$\mathbb{E}\left[T(W)^2\right] = \int\limits_{-t}^{t} w^2 p(w)dw, \qquad (3)$$

where $t$ is the threshold value that controls how much is pruned. Given the compression ratio $c \in (0, 1)$, we find the threshold value which satisfies $P(-t \leq W \leq t) = c$. In case of a symmetric zero-mean distribution, the threshold can be expressed as $t = F_W^{-1}\left(\frac{1}{2} + \frac{c}{2}\right)$, where $F(w) = P(W \leq w)$ is the CDF function and $F^{-1}(p)$ is its inverse. The expected pruning error in equation 3 is similar to the clipping error for quantization (see the second and the third term in equation 2), and can also be computed analytically. We also use this formulation for our results below.

### 3.3 Analytical comparison

**Standard normal distribution.** Let us first look at a standard normal distribution. As many weights in neural networks are roughly Gaussian-shaped, this distribution is useful for our understanding of

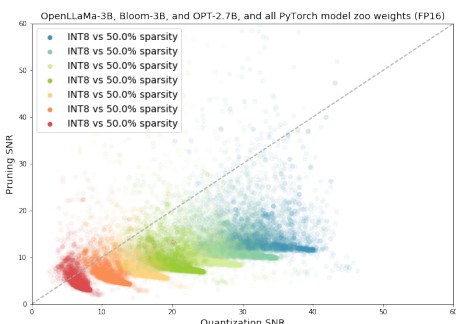
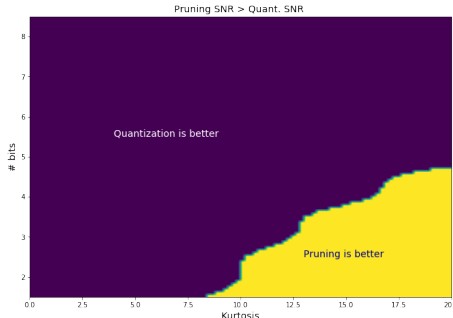

Figure 3: (left) Comparison on all the weights from PyTorch model zoo (46 models) combined with 3 large language models (Bloom-3b, Llama-3b, OPT-2.7b). (left) Pruning SNR versus quantization SNR for every tensor. (right) Pruning is preferable at high compression ratios for tensors with high sample kurtosis values.

the comparison. As we can see from figure 1 (middle), the errors for both methods have very different behavior. The quantization error oscillates between the quantization nodes and has a moderate range. The pruning error effectively corresponds to rounding many weights to zero and thus has a higher error. As we can see in figure 1 (right), this results in a higher SNR for quantization, e.g. 19.1 dB for INT4 quantization versus only 5.6 dB for 75% pruning. We see similar results for different compression ratios. For this distribution, quantization achieves a much higher signal-to-noise ratio.

**Distributions with heavy tails.** The trade-off is expected to change when more significant outliers are introduced. The quantization grid is expected to be effected strongly by outliers as it increases the quantization grid in size, whereas the pruning method is expected to be hardly effected with outliers as it only affects weights around zero. We thus analyze both quantization and pruning errors in the presence of many outliers. To simulate a distribution with outliers, we use a truncated Student's-t distribution with $\nu = 2$, and a symmetric range $(-r, r)$ (the PDF is defined in appendix B). This distribution is nice as it gives a non-trivial weight to the tail ends of the distribution close to $r$. The wider the range $r$ is, the heavier are the tails of the distribution.

In order to introduce a quantitative measure of the number of outliers, we will use the distribution's kurtosis given by $\text{Kurt}[X] = \mathbb{E}\left[(X - \mu)^4\right] / \left(\mathbb{E}\left[(X - \mu)^2\right]\right)^2$, where $\mu$ is the mean. We will see later that this kurtosis measure is predictive of quantiation and pruning performance for real layers. To increase the number of outliers, we will increase the range $r$. The results are given in figure 2. The kurtosis range is chosen so that it includes most of the weights from the model zoo. We see that despite the significant outliers and high kurtosis, quantization still has higher SNR in most of the cases for moderate compression. Pruning is better however in the region of high clipping range and very high compression rate, e.g. 2-3 bits per value (see figure 2 on the right).

### 3.4 Experiments on real weight tensors

The previous discussion was mostly theoretical. We set out to see happens when we do a similar analysis on real neural network weights. In order to investigate this, we compare the pruning and quantization SNR on the weight tensors for all the pre-trained models from the PyTorch model zoo[2] (46 models in total, the details are give in appendix E) combined with weight tensors from 3 large language models, namely Bloom-3b [3], Llama-3b [20], OPT-2.7b [67]. Each tensor is quantized using an integer grid of bit widths from 2 to 8. The results are shown in the figure 3(left). We see a similar trend to our previous discussion that pruning becomes more beneficial for lower bitwidth/higher sparsity ratios.

In order to match the analytical results from figure 2, we consider the sample kurtosis of every weight tensor given by $k = \frac{1}{n}\sum_{i=1}^{n}(x_i - \overline{x})^4 / \left[\frac{1}{n}\sum_{i=1}^{n}(x_i - \overline{x})^2\right]^2$. See figure 3 (right). We consider a range of kurtosis values for every quantization bit-width. Using a kernel density estimator, we compute the probability density of encountering a tensor for which pruning has higher SNR than

---

[2]https://pytorch.org/serve/model_zoo.html.

quantization SNR. We compare the PDF to that for quantization and thus determine the region where each method is preferable. The results are given in figure 3 on the right. We see that the results from the previous theoretical section (figure 2 on the right) hold very nicely. We can also see that as predicted, the kurtosis is indeed a good metric for predicting if a tensor should be quantized or pruned for optimal accuracy.

# 4 Per-layer comparison

Most PTQ methods compress the model layer by layer. Given one layer, we use the mean-squared error of the output activations as an objective for optimization. As [48] shows, minimizing per layer MSE on the output activations of each layer is a computationally affordable second-order approximation of the loss function. The local MSE objective correlates well with the task loss and is often used in practice in DNN compression and quantization literature [32, 40, 68]. Our experiments in appendix D confirm this. For the experiments in this section, we will use SNR as it represents a normalized version of MSE. As opposed to section 3 where we used SNR on weights, in this section, we will use SNR on the output activations instead.

The goal of a PTQ method is to minimize the error in the output activations of the compressed layer by optimizing over the quantized weights subject to integer range constraints. Similarly, for pruning, the weights are optimized subject to a sparsity constraint. As the underlying combinatorial optimization problem for both methods is NP-hard [56, 14], in practice, each method relies on some form of heuristic providing a reasonably good solution given a realistic compute budget. This means that any practical comparison between pruning and quantization would depend on the choice of the method for both and would be open to debate of the optimality of the algorithm. In order to eliminate this dependence, we provide a tight lower bound on the output errors for quantization. For pruning we provide a way to solve the problem exactly for moderate dimensionalities. This way, we can provide a comparison that holds regardless of the algorithm used for each method.

## 4.1 Post-training quantization

We set out to formulate a way by which we can get relatively tight bounds for comparison when quantizing a single layer with the MSE as the objective. The higher bound is simple to obtain by using a solution with a heuristic quantization algorithm, but for the lower bound, we have to reformulate the problem. The mean-squared error of the output activations of a quantized layer can be expressed as:

$$\min_{\boldsymbol{w}} E(\boldsymbol{w}) = \|\boldsymbol{X}\delta\boldsymbol{w} - \boldsymbol{X}\boldsymbol{w}_{orig}\|_2^2 \tag{4}$$
$$\text{s.t. } \boldsymbol{w} \in \mathbb{Z}^n,$$
$$w_{min} \leq w_i \leq w_{max},$$

where $X$ is the input data in an unfolded form, and $w_{orig}$ are the floating point weights. The quantized weights are computed as the product of the quantization scale $\delta$, and the integer weights $\boldsymbol{w}$. $w_{min}$ and $w_{max}$ are the integer limits. We ignore the averaging operation to simplify the notation, as it is not important for optimization. We also note that this problem can be solved independently for each output channel of a convolution or every row of a fully-connected layer weight.

This problem is an instance of a mixed-integer quadratic program:

$$\tilde{E}(\boldsymbol{w}) = \frac{1}{2}\boldsymbol{w}^T\boldsymbol{P}\boldsymbol{w} - \boldsymbol{q}^T\boldsymbol{w}, \tag{5}$$
$$\text{s.t. } \boldsymbol{w} \in \mathbb{Z}^n,$$
$$w_{min} \leq w_i \leq w_{max},$$

where $\boldsymbol{P} = 2\delta^2\boldsymbol{X}^T\boldsymbol{X}$, $\boldsymbol{q} = 2(\boldsymbol{w}_{orig}^T\boldsymbol{X}^T)\boldsymbol{X}\delta$. In order to simplify the objective, we can omit the constant term that is irrelevant for the optimization $c = \|\boldsymbol{X}\boldsymbol{w}_{orig}\|_2^2$, i.e. $\tilde{E}(\boldsymbol{W}) = E(\boldsymbol{W}) - c$.

In order to find the lower bound of the objective, we follow [55] and relax the integer constraint to $w_i(w_i - 1) \geq 0$, which allows the weight to take values within the interval from 0 to 1. In order to

obtain the lower bound, we will consider the dual version of the relaxed problem:

$$L(\boldsymbol{\lambda}) = \max -\gamma, \tag{6}$$
$$\text{s.t.} \begin{bmatrix} \boldsymbol{P} - \mathbf{diag}(\boldsymbol{\lambda}) & q + \frac{1}{2}\lambda \\ \left(q + \frac{1}{2}\lambda\right)^T & \gamma \end{bmatrix} \succeq 0,$$
$$\boldsymbol{\lambda} \geq 0,$$

where $\boldsymbol{\lambda} \in \mathbb{R}^n$, $\gamma \in \mathbb{R}$. The dual problem is convex, and its solution can be used as a lower bound on the solution of the original problem, i.e., $\tilde{E}(\boldsymbol{w}) \geq L(\boldsymbol{\lambda})$. The dual has a semi-definite constraint which can be solved with a semi-definite programming (SDP) solver with $\mathcal{O}(n^3)$ complexity. In our work, we used CVX solver [21]. As discussed in [55], this bound is a computationally efficient alternative to branch-and-bound approaches, while tightness is better than that for the alternative methods introduced in [5]. We use this approach for estimating the lower bound for MSE on the output activations for PTQ below.

## 4.2  Post-training pruning

We also need a similar lower bound for pruning for comparison. To the best of our knowledge we are not aware of the ways to provide a tight lower bound for this problem, therefore we formulate a way to solve a problem for moderate dimensionalities exactly. Similar to quantization, post-training pruning of one layer of the network can mathematically be expressed as solving the following optimization problem:

$$E = \min_{\hat{\boldsymbol{w}}} \|\boldsymbol{X}\hat{\boldsymbol{w}} - \boldsymbol{X}\boldsymbol{w}_{orig}\|_2^2 \tag{7}$$
$$\text{s.t.} \|\hat{\boldsymbol{w}}\|_0 \leq s,$$

where the number of non-zero elements $s$ in the solution is theoretically constrained by using the $L_0$ norm, which is non-convex and not smooth. In order to solve the problem, we introduce the sparsity mask $m \in \mathbb{R}^n$:

$$E(\boldsymbol{w}) = \min_{\boldsymbol{w},\boldsymbol{m}} \|\boldsymbol{X}(\boldsymbol{m} \odot \boldsymbol{w}) - \boldsymbol{X}\boldsymbol{w}_{orig}\|_2^2, \tag{8}$$
$$\text{s.t.} \|\boldsymbol{m}\|_1 = s,$$
$$- \boldsymbol{m} \odot l \leq \hat{\boldsymbol{w}} \leq \boldsymbol{m} \odot u$$
$$l, u > 0, m_i \in \{0, 1\},$$

where $\odot$ is an element-wise product operation, and $l, u \in \mathbb{R}$ are constants chosen such that any solution satisfies the constraint $-\boldsymbol{m} \odot l \leq \hat{\boldsymbol{w}} \leq \boldsymbol{m} \odot u$. We solve this problem using the branch-and-bound method implemented in the Gurobi solver [23] that gives the global solution.

## 4.3  Experiments

With our algorithms in the bag, we can now compare quantization versus pruning in the post-training settings with theoretical bounds. In each case, we analyze individual layers of several networks. Given a batch of input data, we optimize the pruned or quantized weights to minimize the error between the output activations and the output of the uncompressed layer. We provide a range between two SNR values for each method in each case. The performance of the heuristic method gives the first value, and the second value is given by the error lower bound or the global solution, which translates into SNR upper bound.

As a heuristic method for pruning, we use magnitude pruning with a fixed sparsity mask $m$ and data-optimized weights $\boldsymbol{w}$ given by $\boldsymbol{w} = \operatorname*{argmin}_{\boldsymbol{w}} \|\boldsymbol{X}(\boldsymbol{m} \odot \boldsymbol{w}) - \boldsymbol{X}\boldsymbol{w}_{orig}\|_2^2$. This is a convex problem and has a unique solution. As a heuristic method for quantization, we use the mixed-integer solver introduced in [55]. We clip every sample in order to satisfy the integer quantization range constraint.

We chose a representative set of 10 layers, including 9 convolutional layers (one 3x3 convolutional layer and 8 point-wise convolutions) from MobileNet-V2, EfficientNet-lite, and Resnet-18, and one fully-connected layer from ViT. The full details for reproducing the experiments are given in appendix F. Due to the high computational complexity of the global solution for pruning, the layers

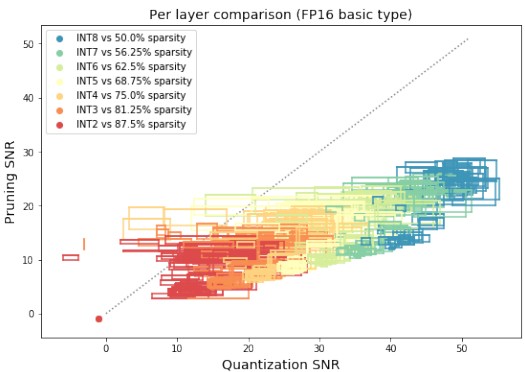

Figure 4: Comparison in the post-training scenario. Each box corresponds to a subset of one of 10 layers from the 4 different models that were used, with 7 different bit-width comparison points. The ranges of the box indicate the lower and higher-bounds found by the algorithms.

had to be split into chunks. The slice of 4 input channels over all output channels was used for 3x3 convolutions. In the case of linear layers and point-wise convolutions, slices 36 input features over all the output features were used.

The results are shown in figure 4 grouped by bit-width. The rectangles indicate the full range of the pruning and quantization methods between the heuristic solution and the error lower bound or the global solution. Whenever a rectangle for each chunk intersects the diagonal line, the ranking of the two methods could depend on the optimization method, while in cases below or above the diagonal, the ranking is guaranteed regardless of the optimizer. We see that quantization mostly outperforms pruning for moderate compression, while methods become more comparable for higher compression ratios.

## 5   Full-model comparison

Now that we have seen the comparison between the methods in the PTQ setting, we turn to fine-tuning quantized and pruned models. This is the setting where pruning is applied in most, and it is possible that fine-tuning can change the models significantly enough that the performance between the two methods changes.

In order to provide a fair comparison of pruning and quantization, we chose the two most commonly used methods with performance competitive to state-of-the-art. For quantization-aware training, we used the widely adapted LSQ method suggested in [12, 2]. Following this approach, we jointly learn the weights and quantization scales, keep the batch norm layers unfolded, and re-estimated the batch norm statistics after training to avoid wrong running estimates due to oscillations [51]. We use the method suggested in [70] for pruning, which gradually increases the sparsity during fine-tuning and re-estimates batch norm statistics after training.

In our experiments we used a set of 4 models trained for 4 tasks including Resnet18, Resnet50 [27], MobileNet-V2 [58], MobileNet-V3-small [30], EfficientNet-lite [60], and ViT [11] trained on Im-ageNet classification [57]; DeepLab-V3 [7] with MobileNet-V2 backbone trained for semantic segmentation on Pascal VOC [13]; EfficientDet [61] trained for object detection on MS COCO [43]; OPT-350 fine-tuned on WikiText-103.

For a fair comparison, we used the same amount of epochs of fine-tuning for each method (full details on hyperparameters are given in appendix G). The results given in table 1 suggest that pruning almost never leads to higher accuracy than quantization if an equal compression rate is considered. The differences are sufficiently large enough that the small purported improvements by some methods [59] will likely not close the gap.

To study the effect of training time, we also performed an ablation with 2 times longer fine-tuning on a subset of 3 models (Resnet50, EfficientNet, and ViT). The results are given in appendix H. We observe that pruned models generally benefit from fine-tuning more, and in particular pruning

| Model | Orig. | Metric | Method | 8b | 7b | 6b | 5b | 4b | 3b | 2b |
|---|---|---|---|---|---|---|---|---|---|---|
| Resnet-18 | 69.7 | acc. | quant. | **70.5** | **70.5** | **70.6** | **70.3** | **70.0** | **68.9** | **67.3** |
| | | | pruning | 70.3 | 70.1 | 69.9 | 69.5 | 69.3 | 68.3 | 66.8 |
| Resnet-50 | 76.1 | acc. | quant. | 76.4 | **76.4** | **76.4** | **76.3** | **76.2** | **75.5** | 72.3 |
| | | | pruning | **76.6** | **76.4** | 76.2 | 76.1 | 75.9 | 75.4 | **74.3** |
| MobileNet-V2 | 71.7 | acc. | quant. | **71.9** | **72.0** | **71.7** | **71.6** | **70.9** | **68.6** | **59.1** |
| | | | pruning | 68.1 | 65.6 | 61.9 | 56.3 | 48.0 | 34.0 | 21.2 |
| EfficientNet | 75.4 | acc. | quant. | **75.2** | **75.3** | **75.0** | **74.6** | **74.0** | **71.5** | **60.9** |
| | | | pruning | 72.5 | 70.9 | 68.1 | 63.6 | 56.4 | 44.5 | 27.1 |
| MobileNet-V3 | 67.4 | acc. | quant. | **67.7** | **67.6** | **67.1** | **66.3** | **64.7** | **60.8** | **50.5** |
| | | | pruning | 65.6 | 64.4 | 62.4 | 60.2 | 56.1 | 31.7 | 0.0 |
| ViT | 81.3 | acc. | quant. | **81.5** | **81.4** | **81.4** | **81.0** | **80.4** | **78.4** | **72.2** |
| | | | pruning | 76.6 | 76.6 | 76.2 | 73.1 | 72.4 | 71.5 | 69.4 |
| DeepLab-V3 | 72.9 | mIoU | quant. | **72.3** | **72.3** | **72.4** | **71.9** | **70.8** | **63.2** | **17.6** |
| | | | pruning | 65.2 | 62.8 | 56.8 | 47.7 | 32.9 | 18.6 | 10.0 |
| EfficientDet | 40.2 | mAP | quant. | **39.6** | **39.6** | **39.6** | **39.2** | **37.8** | **33.5** | **15.5** |
| | | | pruning | 34.5 | 33.0 | 30.9 | 27.9 | 24.2 | 17.9 | 8.0 |
| OPT-350m | 14.8 | perpl. | quant. | **14.8** | **14.8** | **14.9** | **15.0** | **15.3** | **15.9** | **19.9** |
| | | | pruning | 18.0 | 19.7 | 22.6 | 27.2 | 35.4 | 53.5 | 101.4 |

Table 1: Comparison of QAT and magnitude pruning with fine-tuning given equal model size and equal number of epochs for fine-tuning.

becomes more beneficial for most compression ratios on Resnet50. However, for the other models, quantization is still more beneficial due to a larger gap in performance.

**Combining pruning and quantization**     Another interesting questions is whether pruning is beneficial in combination with quantization. To answer it, we perfromed an experiment on pruning quantized Resnet-18, MobileNet-V2 and ViT with different pruning ratios. The results are given on figure 5. On x-axis we plot the expected bit-widths which is a product of the base bit-width and the sparsity in the pruned model including the natural sparsity. The points marked by crosses are quantized models with only natural sparsity and no extra pruning applied. As we can see, mild degrees of pruning are beneficial in the combinations. However, we note that no extra overhead was assumed for storing the pruning mask.

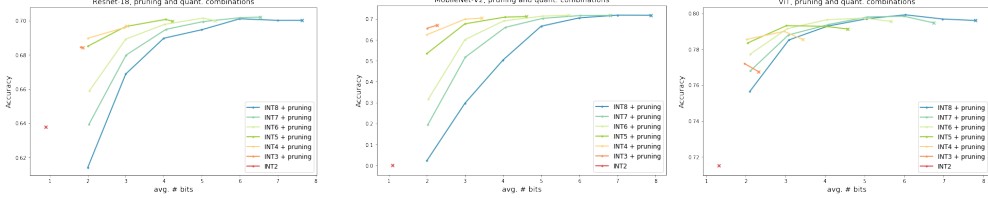

Figure 5: Combining pruning and quantization on ImageNet models. The average bit-widths shown on x axis is computed as a product of the base bit-width and the density of non-zero weight elements. Different pruning ratios are applied to each base bitwidth model. Quantized models with only natural sparsity and no extra pruning are marked with crosses.

## 6  Discussion

**Other types of pruning**     While we solely focused in our comparison on unstructured pruning in which individual weights are removed, our results translate to semi-structured and structured pruning. Unstructured pruning has more degrees of freedom and is a strict superset of what can be

represented by (semi-)structured pruning. Therefore, unstructured pruning gives an upper bound of the accuracy for all pruning methods. This means that for the cases in which quantization is better than unstructured pruning, quantization will also be better than (semi-)structured pruning. However, we can not make any claims for (semi-)structured pruning for the few scenarios in which pruning is better than quantization.

**Natural sparsity in quantized tensors**    In our comparison, we used a theoretical compression ratio for quantization, which depends on the bitwidth. However, we also observe that quantized tensors naturally contain many zeros; for example, 8-bit tensors from PyTorch model zoo have an average sparsity of 13% while 4-bit tensors are 35% sparse. We give more details on this in appendix C.

**Representations learned in the compressed models**    To provide insights into representations learned during pruning or QAT, we studied the evolution of models during fine-tuning. We found that fine-tuning after pruning tends to recover the original representation, while quantization-aware training leads to learning completely new representations. We provide further details on these experiments in appendix I.

**Hardware implications**    So far, we have deliberately avoided discussing the hardware implementations of pruning and quantization and focused solely on the accuracy of both methods at the same ideal compression rates. However, in practice, the hardware considerations do matter for the usability of the methods.

The analysis above assumed an idealistic case for pruning in terms of memory size and data transfer. Since the pruning is unstructured, in order to achieve memory savings in practice, one would need at least 1 bit of information for each weight indicating whether a weight is pruned or not. On top of 16-bit weights, this gives a 6.25% storage overhead at a minimum. Quantization does not have this overhead, as INT8 is just 8 bits smaller than 16 bits, and the only storage overhead is a single scaling factor per tensor (or channel).

Also, in terms of the cost of computations done by the hardware, there is a difference between the two methods. For pruning, any hardware would have to take the densely stored weights and mask and either decompress them to the dense format with all weights and many 0s or take the pruning into account in the compute itself. No compute benefits are gained in the former, as the dense calculations are done in the uncompressed number format. In the latter, dedicated hardware to take into account the 0s is necessary. The overhead for this is generally non-trivial, leading vendors to implement more semi-structured pruning schemes [47]. Similarly, it is rare to see unstructured activation compression for the same reason that this needs to happen algorithmically on-the-fly. In contrast, quantization gives quadratic improvements in the compute. Going from INT8 to INT4 theoretically improves the compute performance by a factor 4, although practical gains depend on the memory overhead (which improves by only a factor 2x) and the existence of other formats in the same hardware compute unit.

**Impact** Using pruning or quantization leads to power reduction on many architectures and enables new applications on mobile platforms. We see only a positive impact from this on the whole. In some cases both pruning and quantization might lead to biased predictions, a further discussion can be found in [29].

**Limitations** First, our work has not extensively considered the hardware implications of pruning or quantization. Second, we do not study combinations of pruning and quantization apart from analyzing the inherent sparsity due to pruning. We leave this for future work. Finally, we consider only uniform quantization and ignore the other formats, such as low-precision floating or logarithmic quantization, although these are not likely to change the results presented in this paper.

# 7    Related work

**Quantization**    Integer quantization, or fixed-point quantization, is one of the most widely used techniques for inference, allowing to reduce the latency and improved energy efficiency. There are two main families of methods for model quantization. The first family includes post-training quantization (PTQ) methods [42, 52, 10, 1, 9, 6, 48, 40], which improve the model accuracy based on per-layer optimization of the quantized weights in a data-optimized fashion. The second family includes quantization-aware training methods [22, 34, 69, 8, 44, 12, 35, 2, 63, 51] which usually fine-tune

the model with quantization in the loop using straight-through estimator (STE) for computing the gradient of rounding operations. A more comprehensive overview of quantization methods can be found in [50].

**Pruning**    Neural network pruning is one of the oldest methods to compress neural networks [37, 26]. A central problem in pruning is how to choose which weights to prune. Approaches published in the literature include: binary gating, in which a binary gate is learned on each individual weight [45, 46, 64]; sensitivity-based methods [39, 38, 66, 17, 18] in which sensitivity, based on a weights' gradient or hessian diagonal value, is used, and magnitude pruning [24, 54, 70, 47, 59]. While conceptually simple, magnitude-based methods have been shown to consistently outperform more intricate methods at scale [19, 4]. Weight re-initialization schemes [15, 16] or mask-reinitialization [59] yield additional minor improvements. While most pruning approaches require fine-tuning and yield unsatisfactory results in post-training scenarios, recent adaptations of Hessian-based sensitivity approaches [37, 26], in which the Hessian of a layerwise reconstruction loss is used instead of the task loss Hessian, show good pruning results in post-training pruning of large language models [17, 18].

**Combining pruning and quantization**    A number of works study combinations of pruning and quantization with different levels of granularity [24, 64, 31, 65, 62, 65].

**Comparing pruning and quantization**    Despite the large amount of work on pruning, quantization, and combining them, there is little literature comparing the two methods. To the best of our knowledge, the closest work that performs a comparison of pruning versus non-uniform quantization  [33]. The work considers only small-scale models and provides only an empirical comparison with no further analysis. Another related study is [53].

## 8    Conclusion

We have seen in this paper that in several settings, unstructured pruning only performs better than quantization in rare cases. In our theoretical analysis of distributions and on the real-layer-data, pruning is only better than quantization, compressing the network to an equivalent of 2 or 3 bits. This amount of compression comes with such a degree of a drop in performance it is rarely used in practice. The post-training quantization results are also informative. In the setting without fine-tuning, we have shown with theoretical bounds on many layers in neural networks that quantization is almost always provably better than pruning. Our hypothesis is that quantized layers are more accurate than pruned ones, as shown in the theoretical and PTQ setting, and fine-tuning a network is still highly dependent on that. This is in line with fine-tuning results, in which for many networks trained under the same conditions, quantization always has higher performance than pruning.

The conclusion is clear: Quantization generally outperforms pruning for neural networks. Taking into account the unfavorable hardware implications for pruning described, it could be argued that the conclusion holds even stronger. Based on this research, we recommend quantizing neural networks when efficiency is required before pruning is explored.

## 9    Acknowledgement

We would like to thank Marios Fournarakis and Yelisei Bondarenko for their help with performing QAT experiments.

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
