## A  Expected quantization error computation

The expected quantization error is a sum of two terms, the rounding error $E_r$ and the clipping error $E_c$:

$$\mathbb{E}(W - Q(W))^2 = E_r + E_c, \tag{9}$$

$$E_r = \int_{q_{min}}^{q_{max}} R_q^2(w)p(w)dw, \tag{10}$$

$$E_c = \int_{-\infty}^{q_{min}} (w - q_{min})^2 p(w)dw + \int_{q_{max}}^{\infty} (q_{max} - w)^2 p(w)dw. \tag{11}$$

The rounding error $E_r$ can be split into two sub-intervals for each interval $(q_i, q_{i+1})$ where the first sub-interval corresponds to rounding up and the second sub-interval corresponds to rounding down:

$$E_r = \sum_{i=1}^{|q|} \int_{q_i}^{q_{i+1}} R^2(w)dw = \sum_{i=1}^{|q|} \int_{q_i}^{(q_i+q_{i+1})/2} (w - q_i)^2 p(w)dw + \\ \sum_{i=1}^{|q|} \int_{(q_i+q_{i+1})/2}^{q_{i+1}} (q_{i+1} - w)^2 p(w)dw. \tag{12}$$

In order to simplify the computation, we introduce the following function:

$$I(a, b, w_0) := \int_a^b (w - w_0)^2 p(w)dw. \tag{13}$$

Thus we can redefine the rounding error as:

$$E_r = \sum_{i=1}^{|q|} \left[ I(q_i, (q_i + q_{i+1})/2, q_i) + I((q_i + q_{i+1})/2, q_{i+1}, q_{i+1}) \right]. \tag{14}$$

We note that the clipping error $E_{cw}$ can also be expressed using $I_w(a, b, w_0)$:

$$E_c = I(w_{min}, q_{min}, q_{min}) + I(q_{max}, w_{max}, q_{max}). \tag{15}$$

where $w_{min}$ and $w_{max}$ are the limits of a truncated distribution. The analytical expressions for $I(w_{min}, q_{min}, q_{min})$ for different distributions are given in the Appendix of [36]. Thus, given the explicit definition of the quantization grid and the probability density function, we can analytically compute the rounding error for different distributions, for example, the Gaussian, Uniform, or Student's t-distribution.

## B  Truncated Student's-t distribution

The PDF of a truncated t-distribution with zero mean and unit variance is given by:

$$f(x, \nu, l) = \frac{p(x, \nu)}{\Phi(l) - \Phi(-l)} \mathbb{1}_{-l \le x \le l}, \tag{16}$$

where $-l$ and $l$ are the truncation limits. $p(x, \nu)$ are the PDF and the CDF of the non-truncated t-distribution given by:

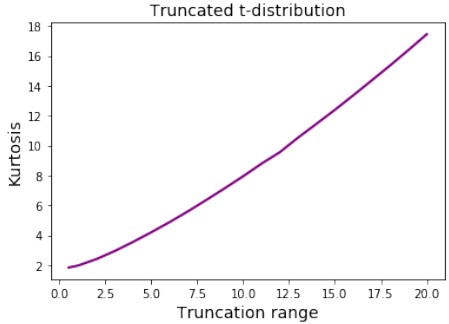

Figure 6: Kurtosis of a symmetric truncated t-distribution as a function of the truncation range.

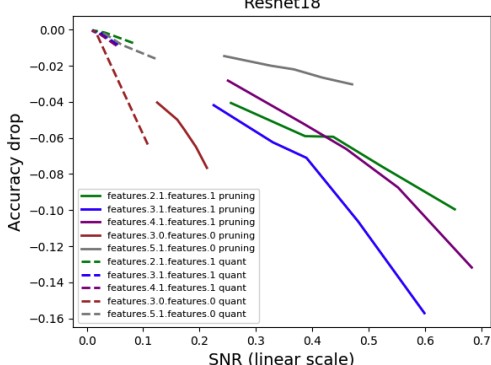

Figure 7: Correlation between per-layer SNR and full-model accuracy for pruning and quantization noise.

$$p(x, \nu) = \frac{1}{\sqrt{\nu}\mathrm{B}\left(\frac{1}{2}, \frac{\nu}{2}\right)} \left(1 + \frac{x^2}{\nu}\right)^{\frac{-\nu+1}{2}},$$

(17)

$$\Phi(x) = \frac{1}{2} + x\Gamma\left(\frac{\nu+1}{2}\right) \times \frac{2\mathrm{F}_1\left(\frac{1}{2}, \frac{\nu+1}{2}; \frac{3}{2}, \frac{-x^2}{\nu}\right)}{\sqrt{\pi\nu}\Gamma\left(\frac{\nu}{2}\right)},$$

(18)

where $F_1$ is the hypergeometric function.

Kurtosis value of this distribution depending on the symmetric truncation range $l$ is plotted in figure 6.

## C    Sparsity in quantized tensors

As we noted in section 6, quantized tensors naturally have some sparsity. The sparsity ratio tends to become higher if lower quantization bit-widths are used. Below we give a table with the average sparsity for all PyTorch model zoo tensors depending on the bit-width:

| num. bits | 8 | 7 | 6 | 5 | 4 | 3 | 2 |
|---|---|---|---|---|---|---|---|
| avg. quant. sparsity | 13% | 16% | 20% | 27% | 35% | 46% | 59% |

Table 2: Natural sparsity in quantized tensors.

As we can see, the sparsity values become very significant, especially for low bit-width values.

## D   Correlation between model accuracy and per layer SNR

In this section, we measure the correlation between SNR at individual layers of a network and the final model accuracy. It is important to study to which degree the two measures are correlated as we used SNR for our experiments in section 4. We note that in this section we use linear scale SNR which is a normalized MSE in contrast to log-scale SNR used in section 4.

The results are given in figure 7. We pruned and quantized single layers of Resnet-18 and plotted activations SNR versus the full model accuracy drop. We observe a strong correlation between SNR and accuracy which confirms our assumption made in section 4.

## E   Details of PyTorch model zoo tensors experiments

We quantized and pruned all the PyTorch model zoo weights tensors. All the convolutional and fully-connected layers were considered, the list is given below (45 models in total).

Classification models:

alexnet, resnet18, resnet34, resnet50, resnet101, resnet152, resnext50-32x4d, resnext101-32x8d, wide-resnet50-2, wide-resnet101-2, vgg11, vgg11-bn, vgg13, vgg13-bn, vgg16, vgg16-bn, vgg19-bn, vgg19, squeezenet1-0, squeezenet1-1, inception-v3, densenet121, densenet169, densenet201, densenet161, googlenet, mobilenet-v2, mobilenet-v3-large, mobilenet-v3-small, mnasnet0-5, mnasnet1-0, shufflenet-v2-x0-5, shufflenet-v2-x1-0.

Object detection models:

fasterrcnn-resnet50-fpn, fasterrcnn-mobilenet-v3-large-320-fpn, fasterrcnn-mobilenet-v3-large-fpn, maskrcnn-resnet50-fpn, keypointrcnn-resnet50-fpn, retinanet-resnet50-fpn, ssd300-vgg16, ssdlite320-mobilenet-v3-large.

Semantic segmentation models:

lraspp-mobilenet-v3-large.

Video classification models:

r3d-18, mc3-18, r2plus1d-18.

## F   Details of per-layer experiments

To reduce the computational complexity of finding the global solution for pruning, the layers had to be split into chunks. The slice of 4 input channels over all output channels was used for 3x3 convolutions. In the case of linear layers and point-wise convolutions, slices 36 input features over all the output features were used.

In section, we provide details on per-layer experiments we performed in section 4. In table 3 we give the names of the models and the layers we used along with the sub-problem dimensionality we considered for each chunk. Depending on the layer, the experiment took from approximately an hour up to six CPU weeks.

## G   Details of the full-model experiments

In our experiments we used a set of 9 models trained for 4 tasks including Resnet18, Resnet50 [27], MobileNet-V2 [57], MobileNet-V3-small [30], EfficientNet-lite [59], and ViT [11] trained on ImageNet classification [56]; DeepLab-V3 [7] with MobileNet-V2 backbone trained for semantic segmentation on Pascal VOC [13]; EfficientDet [60] trained for object detection on MS COCO [43], and OPT-350m fine-tuned on WikiText-103.

In table 4 we provide details of the full-model experiments.

| Model | Layer | sub-problem dim. |
|---|---|---|
| MobileNetV2 | features.8.conv.0.0 | 32 |
| | features.8.conv.1.0 | 32 |
| | features.8.conv.0.0 | 32 |
| | features.11.conv.1.0 | 32 |
| EfficientNet-lite | blocks.1.1.conv_pw | 32 |
| | blocks.6.0.se.conv_expand_pruning | 32 |
| Resnet-18 | layer1.0.conv2 | 36 |
| | layer1.1.conv1 | 36 |
| | layer3.0.downsample.0 | 32 |
| | layer2.0.downsample.0 | 32 |
| ViT | blocks.2.attn.proj | 36 |

Table 3: Details of per-layer experiments.

| Model | Batch size | Weight decay | Optimizer | FT num. epochs | Learning rate |
|---|---|---|---|---|---|
| Resnet-18 | 256 | 1.0e-4 | SGD | 20 | 1.0e-3 |
| Resnet-50 | 128 | 1.0e-4 | SGD | 20 | 1.0e-5 |
| MobileNet-V2 | 128 | 5.0e-5 | SGD | 20 | 1.0e-5 |
| EfficientNet-lite | 128 | 5.0e-5 | SGD | 20 | 1.0e-5 |
| MobileNet-V3 | 128 | 1.0e-4 | SGD | 20 | 1.0e-3 |
| ViT | 128 | 1.0e-4 | Adam | 20 | 1.0e-4 |
| DeepLab-V3 | 16 | 0.0 | SGD | 200 | 1.0e-6 |
| EfficientDet | 16 | 5.0-5 | Adam | 20 | 1.0e-5 |
| OPT-350m | 512 | 0.1 | Adam | 3 | 5.0e-5 |

Table 4: Details of full-model experiments.

As optimal learning for quantization and pruning depends on the compression ratio, we performed a grid search with the step size corresponding to multiplying the basic learning rate above by negative and positive powers of 0.33. For pruning of all the models except for DeepLab-V3 we gradually increase sparsity during the first 15 epochs of fine-tuning and we use the remaining 5 epochs to recover the accuracy with fixed sparsity. A similar scheme is used for DeepLab-V3 with 150 epochs of gradual sparsity increase and 50 remaining epochs of fine-tuning.

For quantization experiments, we use per-tensor quantization and Adam optimizer with a learning rate of 1.0e-5 for quantization scales optimization. We compress weights only and do not prune or quantize activations.

# H  Full-model experiments with longer fine-tuning

In this section we report the results for QAT and magnitude pruning with twice as longer fine-tuning compared to table 4. The results are given in table 5

| Model | Orig. | Metric | Method | 8b | 7b | 6b | 5b | 4b | 3b | 2b |
|---|---|---|---|---|---|---|---|---|---|---|
| Resnet-50 | 76.1 | acc. | quant. | **76.6** | 76.6 | 76.5 | **76.3** | 76.2 | 75.5 | 72.3 |
| | | | pruning | **76.6** | **76.7** | **76.6** | **76.3** | **76.3** | **75.8** | **74.8** |
| EfficientNet | 75.4 | acc. | quant. | **75.2** | **75.3** | **75.0** | **74.6** | **74.0** | **71.8** | **61.5** |
| | | | pruning | 73.0 | 71.9 | 69.5 | 65.8 | 60.0 | 50.7 | 34.6 |
| ViT | 81.3 | acc. | quant. | **81.9** | **81.8** | **81.7** | **81.4** | **80.8** | **78.9** | **73.7** |
| | | | pruning | 80.3 | 79.7 | 79.0 | 77.7 | 76.3 | 74.3 | 71.6 |

Table 5: Comparison of QAT and magnitude pruning with twice as longer fine-tuning compared to table 4.

# I Analysis of representations learned during fine-tuning in QAT and pruning

As fine-tuning significantly improves the accuracy after pruning or quantization, it is interesting to investigate whether fine-tuning recovers the original model. To answer this question, we study how representations at each layer change during the course of fine-tuning by comparing them to the original model representations.

For this purpose, we sample activations from two models, Resnet18 and ViT, after each epoch of fine-tuning, and also directly after quantization/pruning. We measure distances between the original activations and the corresponding activations of the quantized and pruned models. To measure the distance between two feature maps we consider two distance metrics, log-scale SNR and the central kernel alignment (CKA) distance (Kornblith, et al. PMLR, 2019).

We show the results in figure 8. We observe qualitative agreement between both metrics. Curiously, the results show different trends for pruning and quantization. For pruning, the representations tend to become closer to the original representation during fine-tuning. However, for quantization the fine-tuning rather learns a representation that is different from the original one. In order to provide more convenient visualizations, we show one-dimensional plots of both distance metrics at the last non-classifier layer in figure 9 (a-b). We can see that even in the cases of larger distances fine-tuning after pruning tends to recover the representations while it is not the case for QAT.

For ViT we observe qualitatively the same behavior, see figure 9 (c). We omit SNR plots as this measure rapidly becomes negative both for pruning and quantization in ViT. However, CKA evolution follows the same pattern as in the case of Resnet-18 and confirms similar observations.

As we see, fine-tuning during QAT effectively tends to training different representations, while fine-tuning after pruning has a tendency towards recovering the original model.

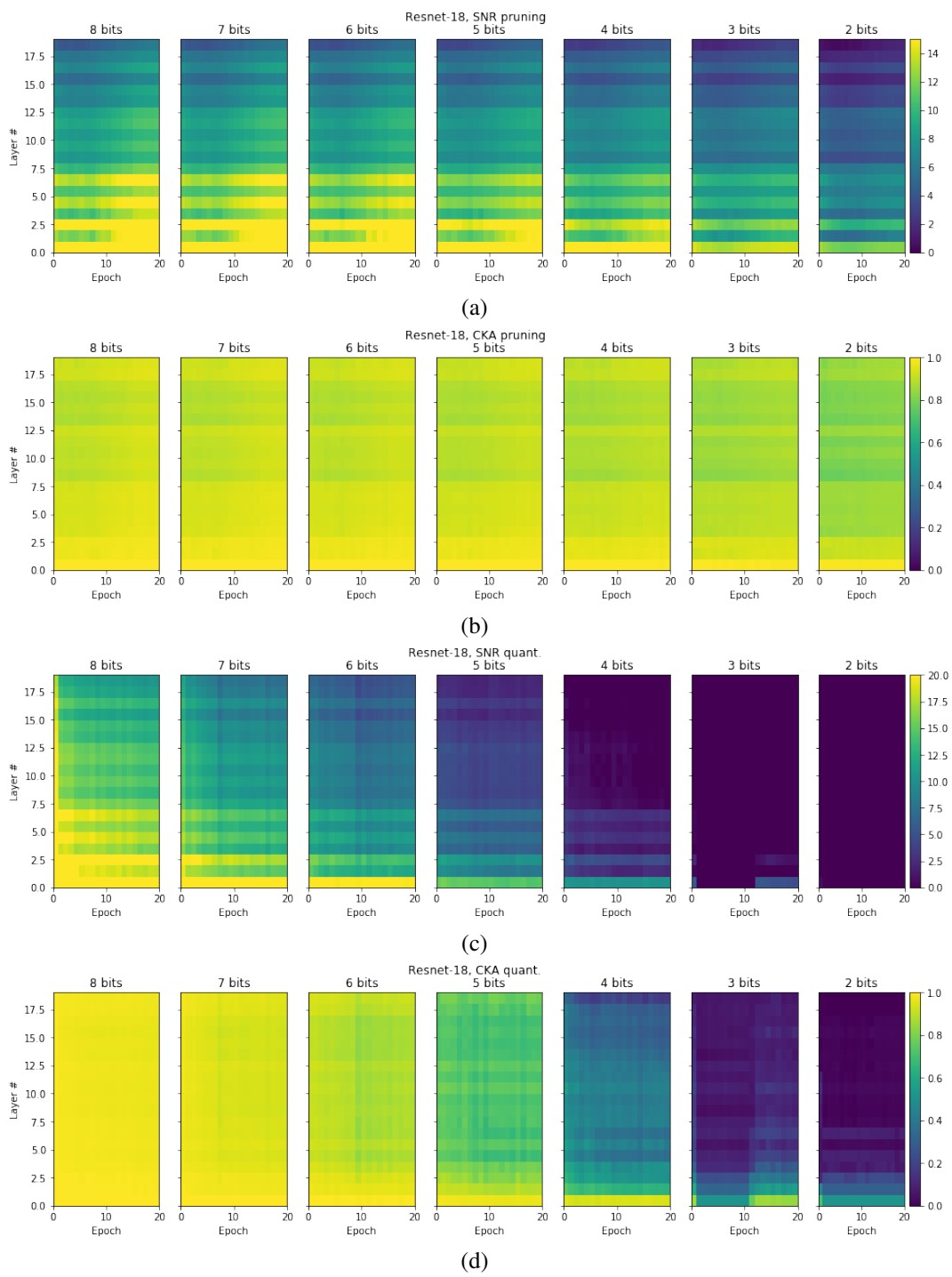

Figure 8: The evolutions of representations at different convolutional layers of Resnet-18 during QAT and fine-tuning after pruning. We plot SNR and CKA distances between activations of each layer of quantized/pruned model and the original activations.

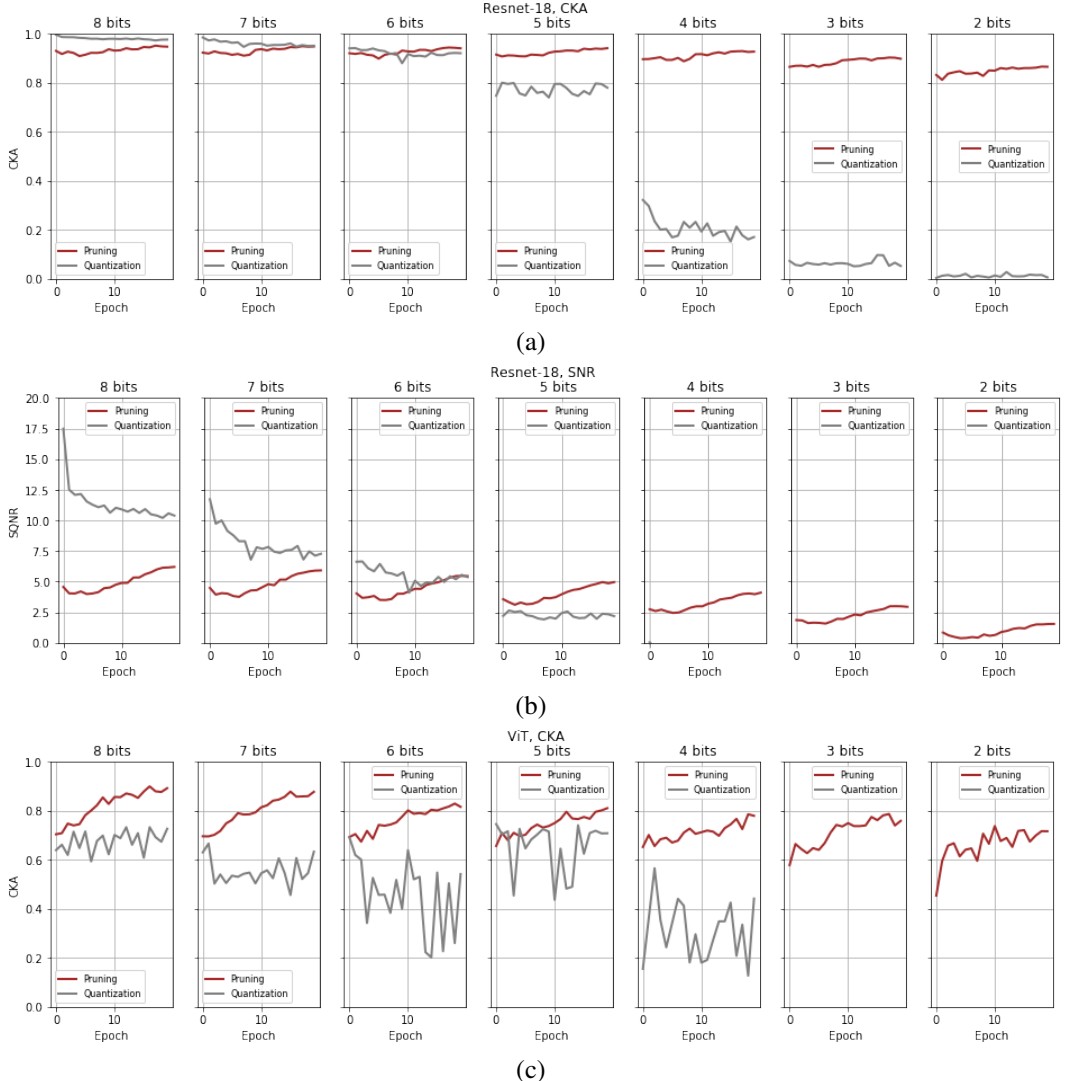

Figure 9: Distances between activations of pruned/quantized models to original activations at the last non-classification layer.