# OpenReview forum: "Pruning vs Quantization: Which is Better?"
_NeurIPS.cc/2023/Conference — NeurIPS 2023 poster_

### Official Review · Reviewer_B8sD · 2023-07-06

**Soundness:** 2 fair
**Presentation:** 2 fair
**Contribution:** 2 fair
**Rating:** 3
**Confidence:** 3

**Summary:**

This submission conducts a series empiricial experiments and anlysis between neural network pruning and quantization. It first used some statistics method to compare pruning/quantization. Then it measure the per-layer error based on a post-training compression framework. Finally, it conducted some experiments using LSQ for quantization and iterative magnitude-based iterative pruning.

**Strengths:**

Statistics method for compression comparison as a non-parameteric and non-trainable method is interesting.

**Weaknesses:**

1. The main drawback of the submission is that it barely proposed new algorithm or insight. The submission is a collection of empiricial experiments and analysis.
2. Comprehensive empiricial experiments can be regarded as contribution. However, the compression methods used in the submission is limited and the experimental setting is not solid. The submission basically focused on magnitude-based pruning and uniform quantization. Author may refer to [1] for some examples.

[1] https://arxiv.org/pdf/1810.05270.pdf

**Questions:**

N.A.

**Limitations:**

N.A.

---

> ### Author Rebuttal · Authors · 2023-08-09
>
> We thank the reviewer for their comments. We answer each of their comments below.
>
> **W1**. We agree that our work does not propose a new quantization or pruning method. However, we respectfully disagree that our paper does not bring new insights. We refer to the general comment on the novelty above for the details on our contribution.  In our view a paper with novel insights is also a valuable contribution. In fact, the reference [1] mentioned by the reviewer is a conference paper which also does not suggest any new pruning method.
>
> **W2**. For our comparison, we used gradual magnitude pruning. First, it is not clear which pruning method exactly is the state-of-the-art. For example, the work Gale et. al 2019 claims that magnitude pruning gives state-of-the art results, or the work [1] suggested by the reviewer also suggests that the choice of the pruning method might not be the key for pruned model accuracy. Regarding magnitude pruning accuracy, in fact we report better results for pruning than the reference [1] which is mentioned by the reviewer. The best Resnet50 ImageNet results are reported in table 6 of [1] from the comment above, for example 76.09% of validation accuracy at 60% sparsity while in our experiments we obtain 76.5% of accuracy at 62.5% of sparsity (see Table 1, Resnet-50 at 6 bits).
>
> We also used uniform quantization for our comparison as a simple and the most widely used technique which is very competitive. Indeed, using non-uniform quantization formats such as FP8 format could lead to slightly better results (see e.g. https://arxiv.org/abs/2208.09225 and https://arxiv.org/abs/2303.17951), however that would only increase the number of cases where quantization is preferrable and thus would not change the overall conclusions of our paper.

---

> > ### Comment · Reviewer_B8sD · 2023-08-17
> > **Response to Authors**
> >
> > I agree that "a paper with novel insights is also a valuable contribution". And my point is that the submission is lacking in insights, due to the following reasons:
> > 1. Only maganitude-based pruning and uniform quantization is compared.
> > 2. The submission can be summary as: (a speific form of) pruning and quantization is compared under same budget and we find that quantization is better in performance.
> >
> > However, for example, pruning (with a same method) can lead to very difference performance under different settings (initialization, training strategy), as shown in [1] (provided by me). Not even to mention quantization. Novel insights do not come from listing numbers but thoughtful comparison and anaysis of the difference.
> >
> > Besides, I don't think reporting a SOTA metric is important and contributes novelty to empricial comparison. Pruning and quantization are two parallel (not mutually exclusive) methods, their performance under same budget can be compared (in complete settings) and indeed important for decision, but their applicability is also very important.
> >
> > Author may focus more on:
> > 1. Thoughtful comparsion (including but not limited to initialization, training, applicability) between pruning and quantization, if the submission's ultimate is still finding which (pruning / quantization) is better in most cases.
> > 2. Add more analysis on statistics methods, and their relation to final performance.
> >
> > Overall, I keep my scores.

---

> > > ### Author Response · Authors · 2023-08-18
> > >
> > > Dear reviewer,
> > >
> > > We believe there might be some misunderstanding and we encourage you to thoroughly read our paper, rebuttal and the papers you cite again.
> > >
> > > We have made it very clear that a change in quantization scheme would not change the outcome of the results in the paper. We also commented that the gradual magnitude-pruning scheme is state-of-the-art or close to it, and many methods that claim to improve over it do so only by small margins. Not sufficiently to refute our results. (See e.g., https://arxiv.org/pdf/1902.09574.pdf or  https://arxiv.org/pdf/2202.01290.pdf).
> > >
> > > The conclusion of the paper you cite yourself [1] is that the pruning methods hardly matter, and networks can be trained from scratch to similar performance. This second part of this paper’s conclusion is in direct contradiction with later and concurrent work such as https://arxiv.org/pdf/2009.08576.pdf,  https://arxiv.org/pdf/1902.09574.pdf. The most  extensive study on pruning methods and their lineage https://proceedings.mlsys.org/paper_files/paper/2020/file/6c44dc73014d66ba49b28d483a8f8b0d-Paper.pdf also came to the conclusion that most differences reported by ‘new pruning methods’ in the literature were most likely due to different experimental setups (which we keep the same in our study). We do not have any reason to believe that the gradual magnitude-based pruning we employ is not state-of-the-art or produces very comparable results to other methods that would negate the conclusions in the paper, especially given the large discrepancy between the pruning and quantization results.
> > >
> > > Our paper does not only list numbers to come to a conclusion. Most of our paper is a very thoughtful comparison, as we compare pruning and quantization theoretically as much as we can, show the difference for either method without data, show the regimes in which one overtakes the other and analyze their performance with algorithmic bounds that give mathematical bounds. We also analyze the training of compressed models and share our insights in Appendix H. The conclusions and findings in the paper stay the same from the theoretical to the practical results, giving strong credibility to our comparisons. We also show the relation between our theoretical and practical analysis on the per-level basis, and the link between the per-layer MSE measures and final performance is well-known and shown again in the paper. Finally, we also provide further evidence for our conclusion in the rebuttal where we generated results for the combination of pruning and quantization – meaning our conclusions likely generalize to this setting as well.
> > >
> > > Finally, our discussion section discusses in-depth on the applicability of the methods, and what would happen in ‘a complete setting’ as you mention.
> > >
> > > If this is not sufficient for you, please indicate technically where you think our reasoning is lacking.

---

### Official Review · Reviewer_XswT · 2023-07-07

**Soundness:** 3 good
**Presentation:** 4 excellent
**Contribution:** 3 good
**Rating:** 7
**Confidence:** 4

**Summary:**

The authors compare the performance of post-training quantization and pruning methods with the same compression ratio using a signal-to-noise metric, a kurtosis metric, and, ultimately, model accuracy. They study the expected performance analytically and in simple toy problems, i.e., Gaussian- and Student's t-distributed weights. They also run experiments on real pre-trained models from the PyTorch Model Zoo. They conclude that post-training quantization alone generally outperforms post-training pruning alone based on their per-layer weight fidelity metrics and full-model accuracy.

**Strengths:**

- The paper represents a complete analysis of post-training pruning alone vs. post-training quantization alone when targeting the same compression ratio.
- The authors analyze the problem at multiple levels (more theoretical/analytical and more empirical/experiment driven) and find the two viewpoints complement and support each other.
- The toy model analysis, though straightforward, illuminates the general issues at play for pruning vs. quantization.


**Weaknesses:**

- The SNR appears to be well-motivated, but it would be ideal to demonstrate explicitly that it maps to model accuracy, e.g. by adding a subfigure with accuracy to Fig. 3 and/or adding the SNR metric to Table 1.
- The authors note that the quantized models naturally exhibit a larger compression ratio than the pruned models because they have a larger fraction of real 0s, which is a minor caveat.
- Some additional references could be added to the related work, e.g.,
  - combining pruning and quantization: https://arxiv.org/abs/2102.11289,
  - quantization-aware training: https://arxiv.org/abs/2006.10159, https://arxiv.org/abs/1905.03696
- In the Impacts section, there is no discussion of the potential decrease in model robustness and generalization when pruning or quantization.

**Questions:**

- Why does the SNR metric use weights and not activations?
- How does the layer-wise SNR metric you evaluate look for the real models in Table 1? Does it match the conclusions based on accuracy?
- Can you indicate how computationally costly the pruning/quantization methods you employ are?

**Limitations:**

The authors discuss the limitations that they do not consider combinations of pruning and quantization, hardware considerations, and other quantization or pruning schemes.

---

> ### Author Rebuttal · Authors · 2023-08-09
>
> We thank the reviewer for the detailed comments and additional references, please find our comments below.
>
> **W1**. The relation between SNR and model accuracy for both pruning and quantization is demonstrated in appendix D (see figure 6).
>
> **W2**. We agree that naturally appearing sparsity in the quantized value make the direct comparison between pruning and quantization more nuanced, however, if we would take the extra zeros into account that would only improve the quantization results and would not change the conclusions of our paper. We refer the reviewer to the general comment on joint pruning and quantization above.
>
> **W3**. We thank the reviewer for the references, and we are happy to include them to the related work.
>
> **W4**. We thank the reviewer for the comment. We would like to add a discussion on additional bias introduced by both compression techniques upon the next revision of the paper.
>
> **Q1**. We use SNR on activations whenever possible (figures 4,5,7,8), while we begin our presentation with weight SNR experiments (figures 1,2,3) in order to build up the study step by step from the distribution analysis to the full model results.
>
> **Q2**. The per layer SNR metric correlates well with accuracy in post-training quantization or pruning as mentioned in figure 6 of appendix D. In this case each compressed layer is optimized based on local SNR objective. However, the SNR layer does not necessarily make sense when the models are fine-tuned. First, there is a list of SNR values corresponding to each layer. Second, when we fine-tune the model we possibly train a new model by minimizing the loss function, and the new model is not constrained to be close to the original model (see appendix H for the related discussion). In this case the SNR metric might not be related to accuracy.
>
> **Q3**. The computational complexity of pruning and quantization techniques we consider is linear with respect to the number of weights. This overhead is relatively low compared to the computational complexity of per-layer optimization (solving the quadratic program) or fine-tuning. In the latter case the straight-through estimator is used for the gradient’s computation so that the computational complexity of fine-tuning is similar to that of the original model training.
>
> **L1. Combining quantization and pruning**. We refer the reviewer to the general comment above for a discussion on combining quantization and pruning.
>
> **L1. Other pruning or quantization schemes**. we do not expect using different quantization or pruning schemes to change the conclusions of our study. For example using 2:4 sparsity or structured sparsity generally performs worse than unstructured sparsity considered in our work (we discuss this in section 6 "other types of pruning"). In our experiment we confirmed that for 50% sparsity when switched from unstructured to 2:4 block sparsity, the validation accuracy decreases for Resnet18 by 0.5%, for Resnet50 by 0.3%, for MNV2 by 1.4%, and for EfficientNet by 1.2%. Using other quantization schemes such as FP8, in its turn, might only improve the results in some cases (https://arxiv.org/abs/2303.17951).

---

> > ### Comment · Reviewer_XswT · 2023-08-21
> >
> > Thank you for the response. I believe some of this information should be included in the paper or supplementary (for example the arguments/studies about why different quantization/pruning schemes won't change the main conclusions). Overall, I stand by my original score.

---

### Official Review · Reviewer_jvrg · 2023-07-07

**Soundness:** 4 excellent
**Presentation:** 3 good
**Contribution:** 3 good
**Rating:** 6
**Confidence:** 4

**Summary:**

This paper sets out to answer the question whether quantization or pruning is better. It first provides an analytical analysis of the two methods in terms of signal-to-noise ratio (SNR) and establishes an early relationship between kurtosis and SNR. It then provides a mathematical breakdown of the compression error in both methods. Then, it empirically compares the two methods on 46 models from the PyTorch model zoo to validate the early analytical results. The authors then compress and fine-tune a set of vision models to compare the two methods after fine-tuning and show that quantization still outperforms in this region. Finally it explores the hardware considerations since dense pruned models can be difficult to accelerate on most hardware.

**Strengths:**

The authors use the reasonable FP16 baseline instead of the more common FP32.

This paper makes an interesting connection between kurtosis and the relative performance of quantization over pruning.

Observation that pruning tends to recover the original representation but quantization builds new ones is intriguing.

The empirical evaluation spans across image classification and detection tasks.

**Weaknesses:**

Especially when considering the hardware, it would be useful to consider block sparsity like 2:4, which is supported in some modern GPUs. However, element-wise sparsity presumably has strictly better accuracy / memory performance, so block sparsity would likely be worse.

Given the recent success of transformers, it may be helpful to include more (in addition to ViT) in the empirical evaluation. That said, I do not expect this will change the conclusion.

You correctly point out that weight distributions are typically quantized symmetrically but they also typically use channel-wise quantization, which is especially helpful with low bitwidths. I expect channel-wise quantization could further shrink the region where pruning is preferred.

I am slightly concerned about the novelty of this comparison, but I will do a better literature search later in the review process. One other potential critique may be that this conclusion may not be too surprising given the exponential decay in the importance of bits. This decay likely does not hold for individual weights.

Minor: L87 is a little confusing for the definition of T, which is more the error of the magnitude pruning, as opposed to the magnitude pruning itself.

**Questions:**

Did you consider clipping to make the distribution less sensitive to outliers? This seems standard with weight quantization too since weights are easy to profile and optimize the quantization settings.

Figure 1 (right) is computed with the analytical equations with the normal distribution substituted?

Are there any regions when the combination of the two techniques outperforms the techniques individually?

I'm confused about Appendix H, where the conclusion is that quantization tends to relearn different representations but pruning recovers the previous ones. The figure shows that the distance between the original and new activations is larger for pruning, which should imply that it doesn't relearn the original distributions.

---

> ### Author Rebuttal · Authors · 2023-08-09
>
> We thank the reviewer for the details review and useful suggestions. Please find our answers and comments below.
>
> **W1**. Block sparsity is a subset of structured sparsity and therefore indeed we expect it to have strictly worse accuracy for the same model size. This was confirmed in our experiments, e.g., for 50% sparsity when switched from unstructured to 2:4 block sparsity, the validation accuracy decreases for Resnet18 by 0.5%, for Resnet50 by 0.3%, for MNV2 by 1.4%, and for EfficientNet by 1.2%. Hence, more structured sparsity does not change the conclusions in our paper. Structured-sparsity would be accuracy-wise worse, at the trade-off of getting a more efficient hardware implementation. We will include this discussion upon the next revision of the paper.
>
> **W2** We totally agree that adding more transformers is beneficial for our empirical comparison. Below we add the results on SQNR values for tensors of 3 large language models (Bloom-3b, Opt-2.7b, OpenLlama-3b). The conclusions are similar to our experiment in figure 3. We are also working on including quantized and pruned Bloom-560m results with fine-tuning. We will include these results to the next revision of the paper.
>
> | Avg. SQNR       | 8b       | 7b       | 6b       | 5b       | 4b       | 3b       | 2b       |
> |--------------|----------|----------|----------|----------|----------|----------|----------|
> | Quantization 	| 33.8 $\pm$ 4.7	| 29.7 $\pm$  4.2 	| 25.7 $\pm$  3.5 	| 21.6 $\pm$  2.8 	| 17.3 $\pm$  2.0 	| 12.7 $\pm$  1.3  	| 7.5 $\pm$  0.8  	|
> | Pruning 	|  12.7 $\pm$  2.3	| 10.8 $\pm$  2.1 	| 9.2 $\pm$  1.7 	| 7.7 $\pm$  1.4 	| 6.3 $\pm$  1.1	 	| 4.9 $\pm$  0.8 	| 3.5 $\pm$  0.7 |
>
> **W3**. We absolutely agree. We avoided using channel-wise quantization through all parts of the study on purpose in order to clearly demonstrate the region where pruning is preferred. For example, if channel-wise quantization was used for 8-bit quantization, the validation accuracy of Resnet18 would increase by 0.2% and MNV2 by 0.3%. Indeed, the region where pruning is preferred would shrink.
>
> **W4**. To the best of our knowledge this comparison has not been presented in the literature, we refer to the general comment above for the details on our contribution. We are eager to cite any relevant work if found.
>
> **W5**. Thank you for the comment, we are going to clarify  the notation in L87.
>
> **Q1**. Yes, we use clipping with the range estimated using MSE error which is a common practice in quantization literature.
>
> **Q2**. Yes, we use the analytical equations for computing the expected error for quantization and pruning, the details are given in appendix A.
>
> **Q3**. A combination of quantization with some mild pruning outperforms quantization to a lower bit-widths in some cases. However, the picture changes if natural sparsity in quantised tensors is taken into account, we refer the comment on combining pruning and quantization above for further details.
>
> **Q4**. The range of CKA distance is from zero to one (the higher the closer), so larger distance for pruning means distributions closer to the original distributions. We kindly ask the reviewer to clarify the question further in case our comment needs further explanation.

---

> > ### Comment · Reviewer_jvrg · 2023-08-18
> >
> > Thank you for addressing my questions. I believe there is value in a large empirical evaluation like this and I'll raise my original score accordingly.

---

### Official Review · Reviewer_S4q6 · 2023-07-09

**Soundness:** 2 fair
**Presentation:** 2 fair
**Contribution:** 2 fair
**Rating:** 4
**Confidence:** 4

**Summary:**

In this paper, the authors try to answer whether pruning or quantization is better for network compression. The paper start by analyze the qunatization error and pruning error under standard normal distribution and heavy-tail distributions. Full-model comparison are done between quantization and pruning with a set of 8 models. In most cases, the quantization method performs better than pruning. Quantization is much better than pruning under extreme compression targets.

**Strengths:**

1. The paper is well-written. The whole writing is to the point. The content is well organized, which makes the paper easy to understand.
2. Thorough full-model comparison is done for a set of 8 models.

**Weaknesses:**

1. The standing point of this paper might be biased. Quantization and pruning are not two competitive methods. Each of them have a specific application scenario. They could be used sololy and jointly. According to the experiments, quantization is almost always better than pruning. Yet, this does not mean that pruning is not useful.
2. The comparison is done such that the model size after pruning and quantization are the same. Not sure whether this comparison is proper.
3. Even without the investigation in this paper, quantization is still used more often than pruning.

**Questions:**

1. Quantization and pruning are two different network compression techniques. But this paper tries to compare them under the setting of same model size. Nevertheless, model size is not the most important factor for efficient applications.

**Limitations:**

The novelty of this paper is quite limited. It is more like an empirical study.

---

> ### Author Rebuttal · Authors · 2023-08-09
>
> We thank the reviewer for comments and suggestions. We answer each point below.
>
> **W1**. We totally agree that pruning is useful, and we did not state the opposite in our paper. Rather, for the setups where both methods are supported, using quantization leads to more accurate models. As we mention in the general comment above, quantization is more accurate even in the joint case.
> For example, pruning is useful in the cases where it is necessary to adjust the model size precisely. The quantised model size is proportional to integer bit-width which makes decrements in the model size relative large. However this can be easily tackled with pruning or a combination of quantization and pruning.
>
> **W2/Q1**. For this work, one might consider at least three metrics: the model size, the bit operations count (BOPs), or run-time on real hardware. The model size is the most simple of the three metrics which is suitable for the direct comparison. In many cases pruning only targets the model size, and resulting memory transfer overheads, unless there are sparse kernels implemented (see section 6 for the discussion on additional pruning overheads). For this reason BOPs count might not be always relevant.
> Finally, as we discuss in section 6 using run-time on real hardware would make the comparison dependent on the specific hardware architecture [mention the discussion]. With that being said, we are open for suggestions on using a different metric if there are some more suitable options.
>
> **W3/Limitation**. We do agree with the reviewer that in practice quantization is used more often than pruning. However, this is in our view not because there are prior extensive and conclusive studies which conclude that quantization is better in terms of accuracy. Likely this is more due to the fact that quantization is more beneficial from a hardware perspective (cf discussion section) and wider supported on common hardware.
> We respectfully disagree that the novelty of our paper is limited and refer to the general comment above.

---

> ### Comment · Senior_Area_Chairs · 2023-08-21
> **final discussions**
>
> Dear Reviewer,
>
> As discussions come to an end soon, this is a polite reminder to engage with the authors in discussion.
> Please note we take note of unresponsive reviewers.
>
> Best regards,
> \
> SAC

---

### Author Rebuttal · Authors · 2023-08-09

We thank the reviewers for their thoughtful comments and useful feedback. We are happy to see that they found the paper well written (S4q6), has a thorough empirical evaluation on various tasks, (S4p6, jvrg), that the comparison is performed on various levels (XswT) and that they found our findings interesting such as attributing quantization/pruning performance to the kurtosis values (jvrg) and the analytical analysis on distributions (B8sD). We address some of the common comments below.
## Combining pruning and quantization
While the original scope of the paper was to compare quantization to pruning, we agree that the paper would greatly benefit from also considering their combinations. Below we would like to share the results on combining pruning and quantization for MobileNet-V2 and Resnet18. Each column corresponds to a compression ratio (measured in average bit-width) which can be achieved using different combinations of pruning and sparsity, e.g. a model quantized to INT6 has size equivalent to INT8 model with 25% sparsity. Each diagonal value corresponds to quantization without pruning. We mark the best combination in each column using bold font.

| MobileNet-V2        | 8b       | 7b       | 6b       | 5b       | 4b       | 3b       | 2b       |
|--------------|----------|----------|----------|----------|----------|----------|----------|
| INT8+pruning         | **71.9** | 71.8     | 59.1     | 47.5     | 35.6     | 0.0      | 0.0      |
| INT6+pruning         | -        | **72.0** | 71.6     | 70.2     | 66.0     | 51.7     | 19.0     |
| INT5+pruning         | -        | -        | **71.8** | 71.4     | 69.1     | 60.1     | 31.7     |
| INT5+pruning | -        | -        | -        | **71.6** | **70.9** | 67.7     | 53.4     |
| INT4+pruning | -        | -        | -        | -        | **70.9** | **70.1** | 64.8     |
| INT3+pruning | -        | -        | -        | -        | -        | 68.6     | **67.9** |
| INT2+pruning | -        | -        | -        | -        | -        | -        | 59.1     |

| Resnet-18        | 8b       | 7b       | 6b       | 5b       | 4b       | 3b       | 2b       |
|--------------|----------|----------|----------|----------|----------|----------|----------|
| INT8+pruning 	| **70.5** 	| 70.3 	| 70.1 	| 69.3 	| 69.0 	| 64.3  	| 61.3  	|
| INT7+pruning 	| -    	| **70.5** 	| 70.2 	| 70.0 	| 69.5 	| 68.0 	| 63.9 	|
| INT6+pruning 	| -    	| -    	| **70.6** 	| 70.1 	| 69.8 	| 68.9 	| 65.9 	|
| INT5+pruning 	| -    	| -    	| -    	| **70.3** 	| **70.0** 	| **69.6** 	| 68.5 	|
| INT4+pruning 	| -    	| -    	| -    	| -    	| **70.0** 	| **69.6** 	| **68.9** 	|
| INT3+pruning 	| -    	| -    	| -    	| -    	| -    	| 68.9 	| 68.4 	|
| INT2+pruning 	| -    	| -    	| -    	| -    	| -    	| -    	| 67.3 	|


As we see from these experiments, quantization alone is better than combinations with pruning in most of cases. Only for a compression ratio below 4 bits, quantization to 3 or 4 bits with added sparsity can be better than the corresponding 2 or 3 bits quantization.

However, even for the low bit cases the picture changes when taking natural sparsity into account (cf. discussion section and appendix C).
Below we give the natural sparsity values for quantization:

| Natural sparsity       | 8b       | 7b       | 6b       | 5b       | 4b       | 3b       | 2b       |
|--------------|----------|----------|----------|----------|----------|----------|----------|
| MobileNet-V2 	| 1.3% 	| 2.4% 	| 4.4% 	| 7.7% 	| 13.4% 	| 23.5%  	| 39.9%  	|
| Resnet18 	|  3.4% 	| 5.3% 	| 8.5% 	| 13.3% 	| 20.3% 	| 31.9% 	| 43.4% |


As we see from the tables in some cases of low bit-width the best accuracy is achieved by combining quantization and mild pruning, for example MN-V2 quantized to INT4 model with 33.33% pruning outperforms INT3 model. However, major part of the reason is natural sparsity in quantized tensors. For example, MN-V2 at INT4 is already 13.4% sparse, while it only needs to be 25% sparse to be compressed down to INT3. This model at INT3 has sparsity of 23.3% out of 33.3% needed to compressed it down to INT2 with pruning.  For Resnet18, the natural sparsity values are higher, i.e. 20.3% at INT4 and 31.9% at INT3, so that the natural sparsity almost achieves what is done with pruning without further accuracy loss. For example, Resnet18 at INT3 pruned down to INT2 has accuracy of 68.4%. If we compare this model to INT3 quantization, the latter has a much higher accuracy of 68.9% which has only 2% fewer zeros. And if we compare INT3 with 33.3% pruning to INT2 model, the latter has sparsity of 43% which makes the model even more compressible.

We further note that pruning only appears beneficial at lower bit-widths where relative overhead of pruning such as storing sparsity mask, which we neglected in this discussion, are potentially much higher (see section 6 of the paper for further details).
We would be happy to include this discussion upon the next revision of the paper.

## Comments on the novelty of the paper, the contribution and its value
Some of the reviewers mention lack of novelty in our paper. We respectfully disagree that the novelty of the paper is limited. To the best of our knowledge the insights from the analytical analysis, the lower bounds for post-training pruning/quantization, and the large scale comparison between pruning and quantization across several tasks were not published before. The closest related work we are aware of is [30] which is limited to small scale empirical comparison. It is important to note that these insights are very valuable for ML practitioners as well as HW engineers making decisions on which methods to support, and where to invest their time when making networks more efficient.

---

### Decision · Program_Chairs · 2023-09-21

**Decision:**

Accept (poster)

**Comment:**

Although the reviewers acknowledged the clarity of the paper, the interest of the study, the extensive evaluation on different models, they expressed concerns about the settings used to compare quantization with pruning, the use of uniform pruning instead of structured pruning, the use of a simple uniform quantization method, the lack of evaluation of pruning and quantization together, and the lack of a new algorithm. The authors feedback addressed some of the concerns, but failed to convince all the reviewers, leaving the paper with two acceptance recommendations and two rejection ones.

As the authors provided some experiments of joint quantization and pruning in the feedback, the main point of contention, is the evaluation of a single pruning strategy and a single quantization one. The argument made by the authors, however, correctly answers this: Their conclusion is that, for a given model size, quantization yields better results than pruning in most cases. This was validated for the simplest possible quantization strategy; using a better one would simply further strengthen this argument. This was evaluated for unstructured pruning, which leaves more freedom than structured pruning, and thus, again, using structured pruning would further strengthen the conclusion. The only remaining question is whether the specific unstructured pruning method used by the authors is sufficient to draw this conclusion. The AC agrees with the reviewers, particularly Reviewer B8sD, that evaluating more unstructured pruning methods would make the message stronger. Nevertheless, the method used by the authors is a popular and effective one, and one can expect only relatively small changes in the numbers if using a different method. As such, the AC believes that this study is of interest to the community in its current form. Nevertheless, the AC urges the authors to clarify the reasons why their conclusions generalize beyond the specific methods that they evaluated.